# A Rumor Detection Method Based on Multimodal Feature Fusion by a Joining Aggregation Structure

**Nanjiang Zhong [1],\*, Guomin Zhou [1], Weijie Ding [2] and Jiawen Zhang [1]**

[1] Department of Computer and Information Security, Zhejiang Police College, Hangzhou 310053, China
[2] Big Data and Cyber Security Research Institute, Zhejiang Police College, Hangzhou 310053, China
\* Correspondence: zhongnanjiang@zjjcxy.cn

**Abstract:** Online rumors spread rapidly through social media, which is a great threat to public safety. Existing solutions are mainly based on content features or propagation structures for rumor detection. However, due to the variety of strategies in creating rumors, only considering certain features cannot achieve good enough detection results. In addition, existing works only consider the rumor propagation structure and ignore the aggregation structures of rumors, which cannot provide enough discriminative features (especially in the early days of rumors, when the structure of the propagation is incomplete). To solve these problems, this paper proposes a rumor detection method with multimodal feature fusion and enhances the feature representation of the rumor propagation network by adding aggregation features. More specifically, we built a graph model of the propagation structure as well as the aggregation structure. Next, by utilizing the BERT pre-training model and the bidirectional graph convolutional network, we captured the features of text content, propagation structure, and aggregation structure, respectively. Finally, the multimodal features were aggregated based on the attention mechanism, and the final result was obtained through the MLP classifier. Experiments on real-world datasets show that our model outperforms state-of-the-art approaches.

**Keywords:** rumor detection; propagation structure; text feature; attention mechanism

## 1. Introduction

Social media platforms are typical places for misinformation to spread. Social media platforms are frequently used by rumor-makers to spread false information. Rumors are spread by rumor mongers to manipulate public events, which can lead to negative consequences. For instance, in politics, rumors have influenced opinions on crucial matters, such as Brexit [1] and the 2016 US presidential election [2]. The "information pandemic" [3] brought on by rumors (in the context of the recent new crown outbreak) has resulted in significant opposition to the suppression of the disease. Therefore, it is crucial to identify and control rumors.

Many research studies have invested in rumor detection. The most direct and efficient approaches involve fact-checking, which involves using known facts to validate the veracity of the news. Known facts come from domain experts, authoritative media, popular science websites, etc. By building a library of known facts, the news to be predicted is searched for in the library, and if there is no similar content in the library, it is considered a rumor. However, there are huge costs involved in building a library of known facts, so the coverage of the library of known facts is often extremely limited [4].

The two primary categories of automated rumor detection techniques, aside from fact-checking-based techniques, are content feature-based methods and propagation structure-based methods. Most content feature-based methods take advantage of textual content, such as user retweets and source tweets [5,6]. To extract text features, the majority of them employ pre-trained models, such as word2vec [7], Glove [8], BERT [9], etc. BERT (bidirectional encoder representation from transformers) is the encoder of a bidirectional

transformer. The model uses two methods, masked LM and next sentence prediction, to capture the representation of words and sentences, respectively. However, methods solely based on text content cannot fit all rumor detection methods, mainly due to the following reasons. Firstly, some rumors are created by imitating the sentence patterns and word patterns of normal text. Secondly, adding real information to rumors makes it more difficult for the classifier to judge.

Recently, researchers found that the propagation structures of rumors and truth are significantly different. According to Vosoughi et al. [10,11], rumors spread more quickly, deeply, and widely than the truth. As a result, recent research studies [12–15] have presented high-level representations from propagation structures to identify rumors. With regard to encoders, deep learning algorithms are widely adopted due to their excellent performances [16–18].

Methods based on propagation structures usually represent the spread of rumors as graph or tree structures, with tweets acting as nodes and the interactions formed by retweets and comments acting as edges. To extract features from the graph structure, existing research studies are mainly divided into two types—RNN-based methods and CNN-based methods. RNN-based techniques can capture sequential features from rumor propagation structures [13], including long short-term memory (LSTM), gated recurrent unit (GRU), and recurrent neural network (RvNN). However, RNN-based approaches ignore the correlation between rumors and instead primarily concentrate on the sequential propagation patterns of rumors. As a result, some researchers use convolutional neural networks (CNNs) to extract the connection features of rumors [15]. However, the global propagation structural relationships of graphs cannot be handled by CNN-based algorithms, although they can capture relevant features of local neighbors [19].

Therefore, some researchers [20–22] have attempted to extract high-dimensional feature representations of rumor propagation structures using graph convolutional networks (GCNs). A GCN is a graph-data version of CNN that can successfully capture global graph features by aggregating node neighborhood data.

However, existing propagation-structure-based methods typically only consider the top-to-bottom propagation of rumors when constructing a graph model, ignoring the bottom-to-top aggregation of public opinions during the rumor-spreading process. Starting with the source tweet, a top-to-bottom propagation network is built between tweets through retweets or comments, as seen in the left part of Figure 1. On the other hand, all comments or retweets are the public's opinions on the previous tweet, and opinions are continuously gathered from the bottom to the top, as shown in the right picture of Figure 1. Existing methods often only consider the top-to-bottom propagation of rumors, which is not comprehensive. By considering the aggregation structure of a rumor, we can capture more discriminative rumor features. Considering the aggregation structure is helpful for the enhancement of rumor features, especially in the early stage of rumor propagation, when there are few tweets in the propagation network.

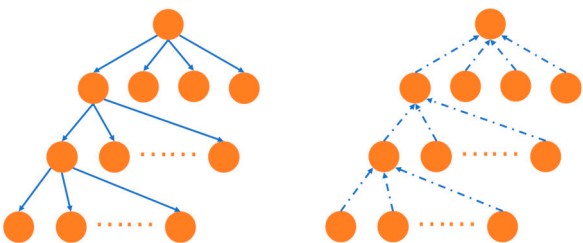

**Figure 1.** The propagation and aggregation of rumors. The orange nodes represent tweets, and the root node is the source tweet. The left part shows the top-to-bottom propagation structures of rumors, and the right part shows the bottom-to-top aggregation structure of public opinions during the spread of rumors.

Additionally, the methods based on the propagation structure cannot effectively utilize the text features of the rumors, even though some existing methods use the text features of tweets as the initialization features of the nodes and then use the graph convolutional network method to extract the propagation structure features. However, the initialized text features gradually fade away during the training process as the iterations increase.

This paper suggests a rumor detection method based on multimodal information fusion that fully considers rumor text features, propagation structure features, and aggregate structure features. First, we used the BERT pre-training model to extract the source tweet text features of the event. Next, we built two graphs for the propagation structure and aggregation structure, and the bidirectional graph convolutional network (Bi-GCN) [21] to extract the propagation and aggregation structure features. Finally, we chose a multilayer perceptron (MLP) to obtain the final result. MLP is a forward-structured artificial neural network. The main contributions of this paper are as follows:

- We introduce an attention mechanism and suggest a multimodal information fusion-based rumor detection method that fully integrates the text features and propagation–aggregation structure features of rumors.
- We improved the rumor propagation network by including the aggregation structure, which can more accurately distinguish rumors from the truth, particularly in the early stages of rumors with an incomplete propagation structure.
- Two real-world Twitter datasets were used in the experiments, and the results show that our method can identify rumors more accurately.

This paper is a continuation of previous work [22] presented at the Fifth ICET conference. The previous version did not discuss the early detection of rumors. This paper supplements the experiments of early detections of rumors, adds related work chapters, and elaborates on the methodology and experimental process.

## 2. Related Work

Rumor detection methods are mainly divided into two categories—content feature-based methods and propagation structure-based methods. The methods based on content features mainly use the text content, picture content, video content, etc., of the original tweet as the model input to detect fake news.

The main method is based on text content. For example, Ma et al. [12] applied deep learning techniques to fake news detection. This method inputs each sentence of the text into the recurrent neural network, uses the hidden layer vector of the recurrent neural network to represent the news information, and inputs the hidden layer features into the classifier to obtain the classification result. Yu et al. [14] used a convolutional neural network to extract text features, and input the obtained embedding vector into the classifier to obtain the final classification result. Vaibhav et al. [23] modeled news articles as graphs with sentences as nodes and inter-sentence similarities as edges; they transformed the fake news detection problem into a graph classification problem. Cheng et al. [24] used a variational autoencoder (VAE) to self-encode text information to obtain the embedding representation of news text, and multi-task learning of the obtained news vector to improve the effectiveness of the model.

News or tweets contain textual as well as visual information, such as pictures and videos. Traditional statistics-based methods use the number of attached images, image popularity, and image type to detect fake news. However, these statistics-based features cannot describe the semantic features of images.

With the rise of deep learning, a large number of works [25–28] are using convolutional neural networks, such as VGG [29] or ResNet [30], to extract features from pictures; researchers are using the extracted features to detect fake news. However, the existing image forgery technology can change the semantic information of the image. The traditional CNN-based model can only extract the pixel-level information of the image, and cannot identify whether the image has been forged.

However, existing rumor makers often write fake news in the same way as real news. Therefore, it is not enough to distinguish fake news based on its content. Sociological studies [10,11] have shown that the propagation of real news and fake news in social networks is often different. Therefore, more researchers are using the propagation structure of news to detect fake news.

For example, Liu et al. [31] regarded the spread of rumors and comment information as a time series; they used RNN and CNN to model the sequence, splice two latent vectors together, and input them into the classification layer to obtain the classification result. Ma et al. [13] modeled the propagation process of rumors as a tree structure. This work constructed a bottom-up propagation tree and a top-down propagation tree, and used recurrent neural networks to extract node features in the tree to classify fake news. Song et al. [32] modeled the news propagation graph as a dynamic graph. Considering the dynamic changes of the news propagation process, the dynamic graph embedding vector was obtained by using the dynamic graph neural network, which was input into the classifier to obtain the classification result.

However, the existing methods did not consider the aggregation features of rumors, which will lead to the loss of information. In addition, the existing methods cannot aggregate the multimodal rumor features well.

### 3. Problem Statement

The definition of the dataset for rumor detection is $C = \{c_1, c_2, \ldots, c_m\}$, where $c_i$ is the $i$-th event and $m$ is the total number of events. $c_i = \left\{r_i, w_1^i, w_2^i, \ldots, w_{n_i-1}^i, G_i\right\}$, where $n_i$ is the number of posts in event $c_i$, $r_i$ is the source post, $w_j^i$ is the $j$-th relevant responsive post, and $G_i = (V_i, E_i)$ is defined as a graph, where $V_i = \{r_i, w_1^i, w_2^i, \ldots, w_{n-1}^i\}$ represents the set of nodes, and $E_i = \{e_{st}^i | s, t = 0, \ldots n_i - 1\}$ represents a forwarding or replying relationship between two nodes.

For instance, there is a directed edge $w_1^i \rightarrow w_2^i$, if $w_2^i$ is a retweet of $w_1^i$, which is represented as $e_{12}^i$. The adjacency matrix is defined as $A_i \in \{0, 1\}^{n_i * n_i}$. $e_{st}^i$ represents the corresponding element value of the $s$-th row and $t$-th column of matrix $A_i$, i.e.:

$$a_{st}^i = \begin{cases} 1, & \text{if } e_{st}^i \in E_i \\ 0, & \text{otherwise} \end{cases}$$

We define $X_i = \left[x_0^i, x_1^i, \ldots, x_{n_i-1}^i\right]$ as the feature matrix composed of all tweets in event $c_i$, where $x_0^i$ is the feature vector of the source tweet $r_i$ and $x_j^i$ is the feature vector of other responded tweets $w_j^i$. In this paper, we used the BERT pre-trained model to extract the feature vector of each tweet's content. Furthermore, a ground-truth label $y_i$ was connected to each event $c_i$. In this paper, $y_i \in \{N, F, T, U\}$, stands for non-rumor, false rumor, true rumor, and unverified rumor, respectively.

The objective of rumor detection is to learn a classifier from the dataset, i.e.:

$$f : C \rightarrow Y$$

where $C$ and $Y$, respectively, stand for the sets of events and labels.

### 4. Materials and Methods

Figure 2 depicts the general workflow of the model employed in this article. We first extracted the text features from tweets using the BERT pre-training model, then we extracted the propagation and aggregation features from rumors using Bi-GCN, and finally we utilized a fully-connected neural network to obtain the result.

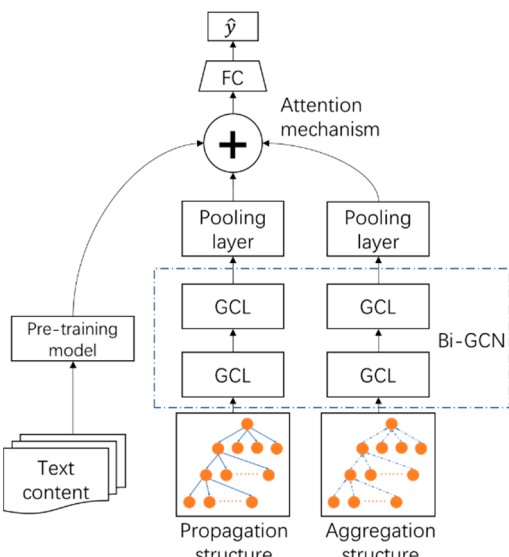

**Figure 2.** Our rumor detection model. We used the BERT pre-training model to extract text content features, extracted the rumor propagation structure and aggregation structure through Bi-GCN, and fused multimodal features through the attention mechanism; finally, the results were obtained by the fully-connected neural network classifier.

We will go into more detail about the model in this section. We exclude the subscript *i* from the following part in order to better illustrate our methodology.

### 4.1. Text Feature Extraction

To extract the content features of rumor texts, the BERT [9] pre-training model was used in this part. BERT can more effectively address the issue of polysemy, i.e., the output of the model for the same word in different contexts is also different (in contrast to the standard word2vec, Glove, and other approaches).

Figure 3 shows the pre-training model architecture of BERT. Each word's embedding, which is divided into three parts by the input layer (token embedding, segment embedding, and position embedding) is one of them. Token embedding is a traditional word-embedding method, such as the one-hot method. There are two embedded special tokens, [CLS] and [SEP], at the start and end of the sentence, respectively. The sentence number to which the word belongs to is labeled using segmentation embeddings. Positional embeddings are used to represent the input sequence's sequential features. The bidirectional transformer allows the BERT pre-training model to learn how the context word affects the current word, which improves the extraction of semantic deep features.

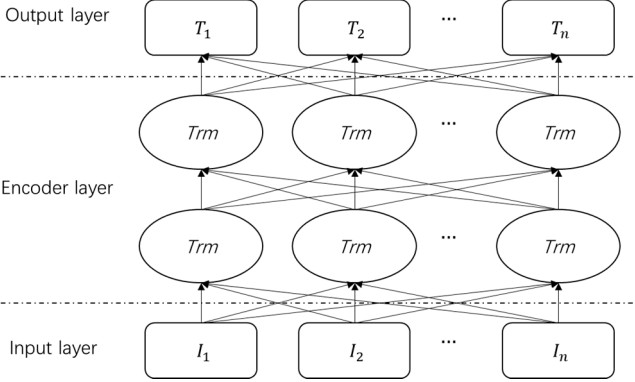

**Figure 3.** BERT model structure.

We utilized the BERT pre-training model to extract the content features of each tweet text in each event, which is represented as $x_0$. We then used an additional two layers of fully-connected layers to convert it into a $v_2$-dimensional vector $S^{TEXT}$, in order to keep its dimensions consistent with the dimensions of the propagation structure feature and aggregation structure feature, which will be introduced later.

### 4.2. Structure Feature Extraction

By using a bidirectional graph convolutional network (Bi-GCN), we extracted event propagation and aggregation structure features. Compared with traditional GCN, Bi-GCN can capture bidirectional neighbor features in the graph model. Based on the retweet and respond relationship, we built a propagation structure graph for each event. Meanwhile, we denoted the adjacency matrix and feature matrix as $A \in \{0,1\}^{n*n}$ and $X$, respectively. $A$ and $X$ were the inputs of the model.

The Bi-GCN was made up of two components: a bottom-to-up graph convolutional network (BU-GCN) and a top-to-down graph convolutional network (TD-GCN). Even though their model structures were relatively similar, they had different adjacency matrices. The adjacency matrix in TD-GCN is represented as $A^{TD} = A$, whereas in BU-GCN, it is represented as $A^{BU} = A^T$, which is the transposition of $A$. The feature matrix $X$ is the same for TD-GCN and BU-GCN.

The features of an event's propagation structure and aggregation structure were extracted based on TD-GCN and BU-GCN, respectively. To extract features in TD-GCN, we utilized two graph convolutional layers (GCL); the calculation formula is as follows:

$$H_1^{TD} = \sigma\left(A^{TD}XW_0^{TD}\right) \tag{1}$$

$$H_2^{TD} = \sigma\left(A^{TD}H_1^{TD}W_1^{TD}\right) \tag{2}$$

where $H_1^{TD} \in \mathbb{R}^{n*v_1}$, $H_2^{TD} \in \mathbb{R}^{n*v_2}$ is the output of the first layer and second layer GCL of TD-GCN, known as the hidden states; the total number of nodes is $n$, the first layer's output dimension is $v_1$, the second layer's output dimension is $v_2$. $W_0^{TD} \in \mathbb{R}^{d*v_1}$ and $W_1^{TD} \in \mathbb{R}^{v_1*v_2}$ are parameter matrices in TD-GCN. For the activation function, $\sigma(\cdot)$, we employed the ReLU function. As shown in (3) and (4), the calculations of $H_1^{BU}$ and $H_2^{BU}$ in BU-GCN are the same as $H_1^{TD}$, $H_2^{TD}$.

$$H_1^{BU} = \sigma\left(A^{BU}XW_0^{BU}\right) \tag{3}$$

$$H_2^{BU} = \sigma\left(A^{BU}H_1^{BU}W_1^{BU}\right) \tag{4}$$

The pooling layer's input was made up of $H_2^{TD}$ and $H_2^{BU}$. The propagation structure feature and aggregation structures were read-out in the pooling layer using the average pooling approach, as illustrated in (5) and (6):

$$S^{TD} = meanpooling\left(H_2^{TD}\right) \tag{5}$$

$$S^{BU} = meanpooling\left(H_2^{BU}\right) \tag{6}$$

The operation of *meanpooling* in (5) and (6) is shown in Figure 4, where the feature matrix was $H_2^{TD}$ or $H_2^{BU}$, a sliding window of size $n \times 1$ was set (shown by the dotted line in Figure 4). The average of the elements in the window was calculated from the first column, and the window was moved backward in turn; finally, a pooling result of size $1 \times v_2$ was made, i.e., the propagation structure feature $S^{TD}$ and the aggregation structure feature $S^{BU}$.

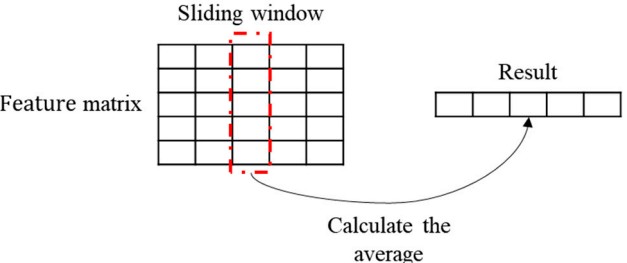

**Figure 4.** The operation of mean-pooling.

### 4.3. Attention Mechanism

The attention mechanism allows the model to focus on the key points, rather than treating them equally. Many sequence-based [33] or image-based [34] tasks have shown the effectiveness of attention mechanisms. In this paper, we faced the multimodal features of text content, propagation structure, and aggregation structure, and we integrated them by introducing an attention mechanism. In the learning procedure, we estimated the attention coefficients adaptively for various types of information. We have:

$$S = \sum_{t \in type} softmax(\alpha_t) \cdot S^t \tag{7}$$

where $type \in \{TEXT, TD, BU\}$ is a set of the feature type; the elements in the set represent the text, propagation structure, and aggregation structure, respectively. Moreover, $softmax(\alpha_t) = \frac{\exp(\alpha_t)}{\sum_i \exp(\alpha_i)}$, where $\alpha_t$ is a free parameter needed to be estimated. The rumor's final feature is $S$.

### 4.4. Rumor Classification and Training

Compared with other classifiers, MLP classifiers have better recognition rates and faster classification speeds. So, the MLP classifier and a SoftMax layer were used to calculate the event's predicted label $\hat{y}$:

$$\hat{y} = Softmax(FC(S)) \tag{8}$$

$\hat{y}$ is a four-dimensional vector, with each element's value representing the likelihood that the event belongs to the corresponding class.

During model training, we minimized the cross-entropy between predictions and ground truth for all events. Additionally, to prevent the overfitting issue, we added the $L_2$ regular term throughout the training process. The definition of the loss function $L$ is:

$$L = \sum_{|C|} \sum_{i \in \{0,1,2,3\}} -y_i log\hat{y}_i + \beta L_2 \tag{9}$$

where $\beta$ is the coefficient of the regular term and $|C|$ is the total number of events. The value $\hat{y}_i$ and $y_i$ are each element of $\hat{y}$ and $y$, representing the probability that the event belonged to the corresponding class.

## 5. Experiment Results

In this section, we show the experimental results of our approaches on two real-world datasets.

### 5.1. Datasets

On two real-world datasets, Twitter15 and Twitter16 [35], we evaluated our suggested methodology. The datasets came from Twitter, one of the most popular social media platforms in the world. We created a propagation graph for each event in both datasets, using nodes to represent tweets and edges to represent their relationships with retweets or comments. The labels of each event in Twitter15 and Twitter16 were annotated according

to the authenticity labels of articles on rumor-debunking websites (such as snopes.com, Emer-gent.info, and so on), whereas Ma et al. [35] refined the labels from binary classes. They have been expanded to quaternary classes: non-rumor (N), false rumor (F), true rumor (T), and unverified rumor (U). Table 1 shows the comprehensive statistics for the two datasets.

**Table 1.** Statistics of the datasets.

| Statistic | Twitter15 | Twitter16 |
| --- | --- | --- |
| # of posts | 331,612 | 204,820 |
| # of events | 1490 | 818 |
| # of True rumors | 374 | 205 |
| # of False rumors | 370 | 205 |
| # of Unverified rumors | 374 | 203 |
| # of Non-rumors | 372 | 205 |
| Avg. # of posts/event | 223 | 251 |
| Max # of posts/event | 1768 | 2765 |
| Min # of posts/event | 55 | 81 |

*5.2. Experimental Settings*

This paper chooses the following baseline models for the comparative experiments:

DTC [36]: A rumor detection approach that employs decision tree as a classifier and obtains credibility based on handcrafted features.

SVM-RBF [37]: A rumor detection method that uses post statistics as features and an SVM model with an RBF kernel for classification.

SVM-TS [38]: A linear SVM classifier used to generate time series models with handcrafted features.

SVM-TK [35]: A rumor propagation structure-based SVM classifier with a propagation tree kernel.

RvNN [13]: A rumor detection approach that uses a tree-structured recurrent neural network with a GRU unit to learn rumor representation via the rumor propagation structure.

PPC_RNN+CNN [31]: A rumor detection model that uses rumor propagation paths to learn rumor representations by combining RNN and CNN.

The features used by each model are quite different; we describe them in detail in Table 2.

**Table 2.** Comparison of the features of all methods.

| Method | Handcrafted Feature | Text Content | Propagation Feature | Aggregation Feature |
| --- | --- | --- | --- | --- |
| DTC | ✓ | - | - | - |
| SVM-RBF | ✓ | - | - | - |
| SVM-TS | ✓ | - | - | - |
| SVM-TK | - | - | ✓ | - |
| RvNN | - | ✓ | ✓ | - |
| PPC_RNN+CNN | - | ✓ | ✓ | - |
| Ours | - | ✓ | ✓ | ✓ |

We separated the two datasets into five parts at random and performed a five-fold cross-validation. The parameters of the model suggested in this study were updated using stochastic gradient descent, and the model was optimized using the Adam method with a learning rate of 0.002. Each node in Bi-GCN had an output dimension of 196 dimensions, which was also the output dimensions of the event text features, propagation structure features, and aggregation structure features. The training process was repeated 200 times, and the training ended when the validation loss stops reduced after 10 iterations.

### 5.3. Confusion Matrix and Metrics

The experiment in this paper is a four-classification problem, and its confusion matrix is shown in Table 3.

**Table 3.** Confusion matrix.

| Ground Truth | Predicted Value | | | | |
|---|---|---|---|---|---|
| | **N** | **F** | **T** | **U** | **Total** |
| N | A11 | A12 | A13 | A14 | N1 |
| F | A21 | A22 | A23 | A24 | N2 |
| T | A31 | A31 | A33 | A34 | N3 |
| U | A41 | A42 | A43 | A44 | N4 |
| Total | M1 | M2 | M3 | M4 | - |

Each value in Table 3 indicates the number of events in a category that are predicted to be a certain value. For example, A12 indicates the number of events with true labels as non-rumors (N), but were predicted to be false rumors (F).

We compare models using accuracy and F1 values of each class. The accuracy refers to the closeness of the predicted value to its "true value", which is a commonly used and an effective evaluation indicator. The calculation formula is as follows:

$$Accuracy = \frac{A11 + A22 + A33 + A44}{N1 + N2 + N3 + N4} \tag{10}$$

The *F1* value is an indicator used in statistics to measure the performance of a classification model. It takes into account both the precision and recall of the classification model.

In the binary classification, the formula for calculating the *F1* value is:

$$Precision = \frac{TP}{TP + FP} \tag{11}$$

$$Recall = \frac{TP}{TP + FN} \tag{12}$$

$$F1 = \frac{2 * (Precision * Recall)}{Precision + Recall} = \frac{TP}{TP + \frac{FN+FP}{2}} \tag{13}$$

where *TP* is true positive, *FP* is false positive, and *FN* is false negative. In the multi-classification problem in this paper, the formula used for calculating the *F1* value of each category can be obtained by a simple derivation. The final *F1* value calculation formula is as follows:

$$N - F1 = \frac{A11}{A11 + \frac{(M1-A11)+(N1-A11)}{2}} \tag{14}$$

$$F - F1 = \frac{A22}{A22 + \frac{(M2-A22)+(N2-A22)}{2}} \tag{15}$$

$$T - F1 = \frac{A33}{A33 + \frac{(M3-A33)+(N3-A33)}{2}} \tag{16}$$

$$U - F1 = \frac{A44}{A44 + \frac{(M4-A44)+(N4-A44)}{2}} \tag{17}$$

### 5.4. Comparative Experiment

All approaches were tested in comparative experiments on the Twitter15 and Twitter16 datasets, and the findings are given in Tables 4 and 5. The experimental results demonstrate that the approach in this work is superior to other baseline methods, with accuracies of 83.6% and 85.1%, respectively, and the influence on the F1 value is also superior among

these models. It shows that our method can effectively distinguish rumor and truth by aggregating multimodal features.

**Table 4.** Rumor detection results on Twitter15.

| Method | ACC. | N | F | T | U |
|---|---|---|---|---|---|
| | | F1 | F1 | F1 | F1 |
| DTC | 0.452 | 0.414 | 0.365 | 0.715 | 0.332 |
| SVM-RBF | 0.301 | 0.214 | 0.158 | 0.481 | 0.239 |
| SVM-TS | 0.537 | 0.781 | 0.476 | 0.395 | 0.462 |
| SVM-TK | 0.745 | 0.794 | 0.651 | 0.750 | 0.742 |
| RvNN | 0.741 | 0.691 | 0.773 | 0.817 | 0.630 |
| PPC_RNN+CNN | 0.485 | 0.357 | 0.513 | 0.372 | 0.712 |
| **Ours** | **0.836** | **0.840** | **0.805** | **0.903** | **0.831** |

**Table 5.** Rumor detection results on Twitter16.

| Method | ACC. | N | F | T | U |
|---|---|---|---|---|---|
| | | F1 | F1 | F1 | F1 |
| DTC | 0.491 | 0.316 | 0.209 | 0.460 | 0.591 |
| SVM-RBF | 0.633 | 0.680 | 0.106 | 0.184 | 0.381 |
| SVM-TS | 0.615 | 0.767 | 0.437 | 0.560 | 0.544 |
| SVM-TK | 0.738 | 0.718 | 0.775 | 0.856 | 0.725 |
| RvNN | 0.728 | 0.634 | 0.692 | 0.820 | 0.731 |
| PPC_RNN+CNN | 0.531 | 0.576 | 0.536 | 0.385 | 0.683 |
| **Ours** | **0.851** | **0.810** | **0.844** | **0.917** | **0.856** |

Furthermore, we can see that models SVM-TK and RvNN, which considered event propagation structural features, performed significantly better than other methods that ignored propagation structures, indicating that the propagation structures of rumor events are indeed different from those of ordinary events and are sufficient in the process of rumor detection. Considering the propagation structures of rumors helps to improve detection accuracy.

In addition to traditional text content features and propagation structure features, our method also extracts aggregation structure features. The experimental results show that adding aggregation structure features can detect rumors more accurately.

### 5.5. Ablation Study

We conducted an ablation study to see whether each module contributed to the model and which module contribute more. In other words, we demonstrate that text features, propagation structure features, and aggregation structure features can all benefit from rumor identification. Our proposed model's main modules are BERT, TD-GCN, and BU-GCN, which stand for extract text features, propagation structure features, and aggregation structure features, respectively. Based on the whole model, we systematically removed the above modules and compared their changes in accuracy and the F1 value for each category. Figures 5 and 6 show the experiment results.

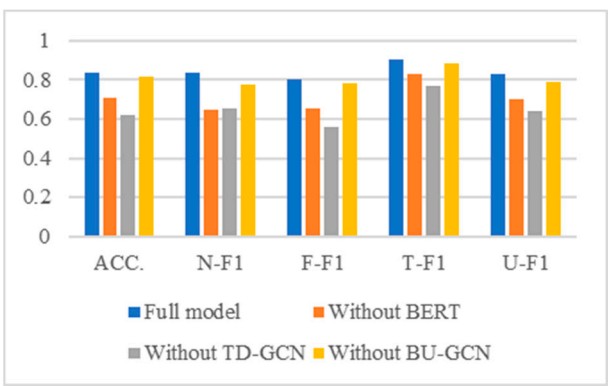

**Figure 5.** Results of the ablation experiments on Twitter15.

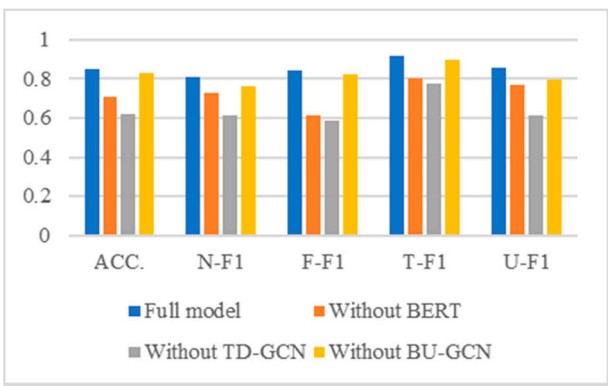

**Figure 6.** Results of the ablation experiments on Twitter16.

The following conclusions can be drawn from the results of the ablation experiments shown in Figures 5 and 6. After removing any module, all indicators decreased, demonstrating that each module in our model has a beneficial effect on rumor recognition results, suggesting that each feature is essential. The indicators declined the most after removing the TD-GCN module, showing that the propagation structure feature had the biggest impact on rumor detection.

The contribution ranking of the three modules in the model for the rumor detection results was: TD-GCN, BERT, and BU-GCN. That is to say, the contributions of different types of features to the results of rumor recognition are ranked from high to low as text feature, propagation structure feature, and aggregation structure feature.

*5.6. Early Detection*

We found that the early detection of rumors is significantly improved by fusing aggregated structures. Fake news is easily forwarded and spread by a large number of users on social media, causing serious impacts in a short period of time. Therefore, the earlier it is detected, the more the negative impacts can be avoided. However, in the early days of rumor spreading, the available spread of data is limited. In order to verify that the method in this paper improved the effect of the early detection of rumors, we used a small number of early rumor data, as shown in Figure 7, to compare the two methods: (1) The method using only the propagation structure (PS); (2) the method to fuse the propagation and aggregation structures (PS+AS). The results are shown in Tables 6 and 7. The ratio column in the table indicates the ratio of the propagation structure used during training.

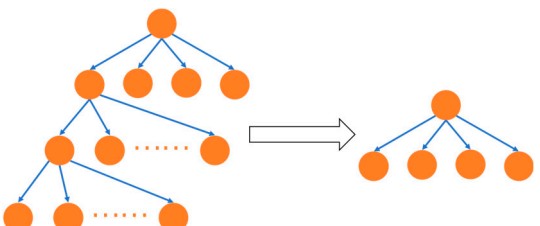

**Figure 7.** Comparison of the early and complete propagation structures of rumors.

**Table 6.** Early detection results of rumors on Twitter15.

| Ratio | Method | ACC. | N | F | T | U |
|---|---|---|---|---|---|---|
| | | | F1 | F1 | F1 | F1 |
| 10% | PS | 0.616 | 0.619 | 0.598 | 0.684 | 0.607 |
| | PS+AS | 0.651 | 0.664 | 0.638 | 0.739 | 0.658 |
| 30% | PS | 0.682 | 0.705 | 0.641 | 0.769 | 0.706 |
| | PS+AS | 0.715 | 0.730 | 0.685 | 0.792 | 0.722 |
| 50% | PS | 0.751 | 0.769 | 0.723 | 0.816 | 0.721 |
| | PS+AS | 0.780 | 0.802 | 0.731 | 0.858 | 0.774 |
| 60% | PS | 0.773 | 0.782 | 0.745 | 0.860 | 0.765 |
| | PS+AS | 0.791 | 0.812 | 0.767 | 0.876 | 0.799 |
| 80% | PS | 0.804 | 0.817 | 0.753 | 0.882 | 0.786 |
| | PS+AS | 0.822 | 0.838 | 0.774 | 0.891 | 0.815 |

**Table 7.** Early detection results of rumors on Twitter16.

| Ratio | Method | ACC. | N | F | T | U |
|---|---|---|---|---|---|---|
| | | | F1 | F1 | F1 | F1 |
| 10% | PS | 0.620 | 0.606 | 0.614 | 0.691 | 0.628 |
| | PS+AS | 0.674 | 0.651 | 0.673 | 0.747 | 0.663 |
| 30% | PS | 0.708 | 0.662 | 0.681 | 0.763 | 0.674 |
| | PS+AS | 0.753 | 0.736 | 0.728 | 0.802 | 0.721 |
| 50% | PS | 0.771 | 0.731 | 0.745 | 0.807 | 0.726 |
| | PS+AS | 0.802 | 0.743 | 0.760 | 0.837 | 0.758 |
| 60% | PS | 0.795 | 0.743 | 0.759 | 0.858 | 0.766 |
| | PS+AS | 0.820 | 0.762 | 0.796 | 0.870 | 0.790 |
| 80% | PS | 0.837 | 0.785 | 0.820 | 0.890 | 0.819 |
| | PS+AS | 0.847 | 0.806 | 0.831 | 0.910 | 0.831 |

From the experimental results of early detection, it can be seen that the method considering the aggregated structure is significantly better than the method that only considers the propagation structure. Moreover, the lower the ratio, the more obvious the performance differences. This shows that considering the aggregation structure can effectively enhance rumor features and improve the distinction between real information and rumor features.

### 5.7. Classifier

In this paper, a fully-connected neural network with four layers was chosen as the classifier, and the model structure was $192 \times 96 \times 48 \times 4$. In this section, we evaluate the effects of different model topologies on the performance of rumor detection through experiments. Layers 2, 3, 5, and 6 had the following model structures, as illustrated in Figure 8: $192 \times 4$, $192 \times 96 \times 4$, $192 \times 96 \times 48 \times 24 \times 4$, $192 \times 96 \times 48 \times 24 \times 12 \times 4$.

Each evaluation indicator was the most effective when the model structure had four layers. This is because when the model structure is too simple, the model under-fits, and when the model structure is too complex, the model slightly over-fits. All of them contributed to a decline in the model performance.

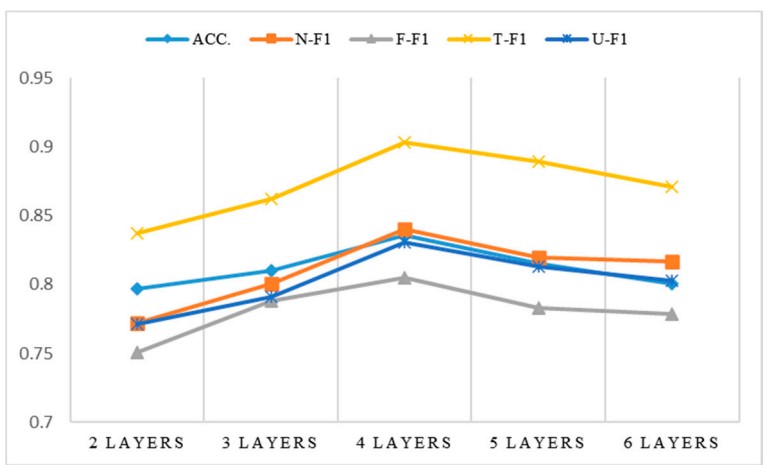

**Figure 8.** The effects of different classifier structures on the results.

## 6. Conclusions

In this paper, we propose a new rumor detection method and compared it with existing methods; we fully considered the aggregation structures of rumors when extracting rumor features. Moreover, through the attention mechanism, the content features of rumor text, propagation structure, and aggregation structure were integrated. More discriminative rumor features were obtained by this method. Among them, we used the BERT pre-training model to extract text features, TD-GCN to extract propagation structural features, and BU-GCN to extract aggregation structural features. The comparative experimental results show that the proposed method can detect rumors more accurately than the existing methods. The ablation study results show that each module of our method has a positive effect on the detection results. The experiments of early detection show that the method considering the aggregation structure can deal with the early detection of rumors more effectively.

Rumor detection is currently a research hotspot in academia, and there are still many problems that need to be solved, for example, (1) how to detect false information in visual information (considering that social media tweets contain a large number of visual information); (2) how to accurately detect rumors in the early stages of a rumor (so as to cut off the spread of the rumor as soon as possible); (3) how to establish an effective mechanism so that the model can provide reasons for making judgments while giving the rumor detection results (so that the detection results are more convincing). In future work, we will conduct more in-depth research on the above issues.

**Author Contributions:** Methodology, W.D. and G.Z.; software, N.Z.; validation, N.Z. and J.Z.; writing—original draft preparation, N.Z.; writing—review and editing, N.Z. and G.Z. All authors have read and agreed to the published version of the manuscript.

**Funding:** This research was funded by the National Natural Science Foundation of China (no. U1509219), Basic Research Project of the Ministry of Public Security (No. 2019GABJC36), and the Youth Foundation of Zhejiang Police College (No. 2021QNY002).

**Institutional Review Board Statement:** Not applicable.

**Informed Consent Statement:** Not applicable.

**Data Availability Statement:** The data included in this study are available upon request by contacting the first author.

**Conflicts of Interest:** The authors declare no conflict of interest.

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
