# Peer review of "A Rumor Detection Method Based on Multimodal Feature Fusion by a Joining Aggregation Structure"

_electronics, doi:10.3390/electronics11193200_

Round 1

Reviewer 1 Report

Dear Authors:

I have enjoyed reading the paper very much. I think it is a work of high social value and highly relevant. I consider the mathematical foundation to be adequate and correct. I consider that the application carried out as an experiment seems very interesting. However, I miss working with a real example that can be easily identified. I do not know if this would be possible. I congratulate the authors for the work. If you can give an answer to this question it would be appreciated.
Congratulations.

Author Response

Dear reviewer,

Thank you for this valuable feedback. We have responded to the questions you asked in the attachment. So, please see the attachment.

Thank you.

Reviewer 2 Report

This paper presents a rumor detection method based on multimodal feature fusion by adding aggregation features, which enhances feature representation. Rumor detection using neural networks is an attractive and significant topic for social media research. This manuscript was well written, and the presentation was well constructed. Please consider the following comments to improve the article.

1. What kind of data set did the authors choose? Please add and introduce the data set more clearly because the result may change by changing the dataset. For example, the authors selected twitter15 and twitter16 for the previous research. The reviewer thinks it will help the reader understand the property of the data set if the authors can provide more information about the character of the dataset issue.

2. The authors only demonstrate an image diagram, such as figures 2 and 3, to show the adopted network model. Then, the authors can add more information to the figures regarding the detailed structure. For example, the dimensions of hidden layers. Moreover, please try to mark the important Mathematical symbols defined in the Problem Statement.

3. No limitations to the proposed method. The authors could discuss the limitations regarding the method corresponding to the training set or the network structure.

4. There is a minor question that BERT has been used as an abbreviation in the abstract.

Author Response

Dear reviewer,

Thank you very much for the positive comments and constructive suggestions. We have made changes to the original manuscript with your comments. For details, please see the attachment. Thank you!

Mr. Zhong

Reviewer 3 Report

The current study represents A Rumor Detection Method Based On Multimodal Feature Fusion. The topic is quite interesting and novel. However, I have found few issues which are given below.

·        Revise your paper for incomplete sentences, such as “In 11 additions, only considering the rumor propagation structure and ignoring the aggregation structure 12 of rumors cannot obtain enough discriminative features. Especially in the early days of rumors, 13 when the structure of the propagation is incomplete”.

·         MLP classifier and BERT pre-training model is not explained before citing. If the author is using these models, then they should provide a brief definition.

·         The authors should mention why they are using MLP, BERT pre-training model and BiGCN.

·         Why figure 01 in introduction? The authors should provide a concise motivation and challenges in the first paragraph of the introduction.

·         Here the author claim “With the rise of deep learning, a large number of works[25][26][27][28] use convolutional neural network VGG[29] or ResNet[30] to extract features from pictures, and use
the extracted features to detect fake news. However, the existing image forgery technology can change the semantic information of the image”. How the proposed model will address this issue?

·         The related work section shows multiple stories of the exiting studies. It is highly recommended to provide comparison based literature review where the future reader can learn about the significance of the current study.

·         Line 168 to line 171, there are multiple phrases that are incomplete.

·         Provide brief explanation in the caption of figure 02.

·         Revise the whole manuscript for typos and grammatical issues.

·         Equation 11 to equation 14 is not well explained. Revise this section as well.

·     I have not found limitations and comparison analysis. Include a separate section for comparison with at least 3 exiting models.

Author Response

Dear reviewer,

Thank you for your kind suggestion about our work. We have responded to each suggestion one by one and made corresponding revisions to the original manuscript. For details, please see the attachment.

Mr. Zhong
